# Mare Milk and Foal Plasma Fatty Acid Composition in Foals Born to Mares Fed Either Flax or Fish Oil During Late Gestation

**DOI:** 10.3390/ani15111612

**Published:** 2025-05-30

**Authors:** Erica A. Snyder-Peterson, Nichola Shost, Timber Thomson-Parker, Kayla C. Mowry, Kalley K. Fikes, Rachelle Smith, Benjamin Corl, Ashley Wagner, Ivan Girard, Jessica K. Suagee-Bedore

**Affiliations:** 1School of Agricultural Sciences, Sam Houston State University, Huntsville, TX 77241, USA; ea.snyder18@gmail.com (E.A.S.-P.); shostnichola@gmail.com (N.S.); parkertimber26@gmail.com (T.T.-P.); kcmowry@yahoo.com (K.C.M.); kkf007@shsu.edu (K.K.F.); 2TRIES LAB, Sam Houston State University, Huntsville, TX 77241, USA; env_rxs@shsu.edu; 3School of Animal Sciences, Virginia Tech, Blacksburg, VA 24060, USA; bcorl@vt.edu; 4Probiotech International, St. Hyacinthe, QC J2S 8L2, Canada; ashleywagner@probiotech.com (A.W.); ivangirard@probiotech.com (I.G.)

**Keywords:** 20:5 n-3, 22:6 n-3, docosahexaenoic acid, eicosapentaenoic acid, late gestation, neonatal foal, omega-3 fatty acids

## Abstract

Fish oil provides animals with eicosapentaenoic (EPA) and docosahexaenoic (DHA) acids, omega-3 fatty acids that have many important roles in animal health. In comparison, terrestrial plants provide the omega-3 fatty acid alpha-linolenic acid, a fatty acid that grazing animals can metabolize into EPA and DHA. The current study supplemented late-gestation mares with one of three treatments: fish oil, flaxseed, or unsupplemented controls. This was carried out to compare the effects and benefits of these fatty acid sources on their blood and milk fatty acid compositions. The mares received treatments beginning on day 310 of gestation and continued through day 5 post-parturition. Blood and milk samples were collected from the mares, and blood samples were collected from the foals. The plasma, harvested from the blood samples, and milk were analyzed for their fatty acid compositions using gas chromatography. Foals born to mares provided with fish oil exhibited increased plasma DHA levels at birth. Regardless of the treatment, the plasma DHA content was higher in foals prior to nursing than on days 5 and 30. No treatment differences were observed in the mares’ plasma or milk fatty acid composition. The most efficient avenue for increasing neonatal foal DHA levels may be through maternal supplementation with fish oils during late gestation.

## 1. Introduction

Essential fatty acids (EFAs) are a specific type of unsaturated fatty acid composed of a long hydrocarbon chain that contains a final double bond between the third and fourth (omega-3) or sixth and seventh (omega-6) carbons from the methyl end. Additionally, there are omega-9 fatty acids that contain a final double bond between the 9th and 10th carbons from the methyl end. While mammals are able to synthesize double bonds between the 9th and 10th carbons, they likely do not produce enough omega-9 fatty acids to meet daily requirements [1]. Furthermore, mammals are unable to produce double bonds closer to the methyl end and, therefore, must consume fats that contain them. This study investigated the compositional changes in omega-3, -6, and -9 fatty acids in the blood and plasma in response to dietary treatment with sources of EFA in late gestation. These EFAs provide a variety of functions in mammalian bodies; for instance, omega-3 fats have roles in nerve cell signal transmission, resistance to mechanical stress, and the anti-inflammatory aspects of responding to cellular injury and infection [2]. In particular, the omega-3 fatty acid docosahexaenoic acid (DHA; 22:6 n-3) is the substrate for anti-inflammatory compounds of the prostaglandin cascade and is, therefore, critical to the body’s repair response to infection and injury [3]. In opposition to this, omega-6 EFAs are critical for signaling pain and initiating the inflammatory cascade, an essential early response to infection or injury [4]. Omega-9 fatty acids also exhibit anti-inflammatory and anti-cancer functions [5].

Horses obtain EFAs through their native diet as well as from dietary supplements. Essential fatty acids are abundant in forages such as grass, hay, and pasture, with these sources having the omega-3 EFA alpha-linolenic acid (ALA; 18:3 n-3) but nearly undetectable eicosapentaenoic acid (EPA; 20:5 n-3) and DHA levels. Additional sources of ALA include oils from seeds such as flax, soybean, and chia, which comprise 55%, 8%, and 64% ALA, respectively. The predominant omega-3 fatty acid consumed by horses is ALA, which can be desaturated and elongated into EPA and DHA through an intracellular process [6], although the efficiency of this process is unknown in horses. The conversion rate efficiencies of ALA to DHA for tree swallows were found to be 3 and 11 percent in the muscles and liver, respectively [7]. In humans and swine, the conversion rates are estimated to be less than 9 percent [8] and are widely affected by the dose, biological sex, and ratio of intake to omega-6 fatty acids [9,10]. In fact, the main role of dietary ALA may be beta-oxidation [10,11]. In swine, increasing muscle EPA levels by 1% required a dietary intake of either 16 g of ALA/kg feed or 3–5 g of EPA/kg feed [10], suggesting that increasing dietary ALA levels may not be the most efficient approach to increase the amount of EPA and DHA in tissues. Dietary supplementation with DHA and EPA, typically found in fish and marine algal oils, may be a more effective way to directly supply tissues with these fatty acids.

Terrestrial grasses produce the EFAs ALA and linoleic acid (LA; 18:2 n-6) in a ratio that ranges from 1.6:1 to 3.6:1 [12]. As such, the native grass-based diet of equids contains negligible EPA and DHA levels, which suggests that the conversion rates of ALA to these longer-chain omega-3 fatty acids are sufficient to meet individual needs. From other species data, it is also known that the conversion rate may be influenced by both the total intake and the ratio of intake of omega-3 to omega-6 [10]. One of the reasons to supplement domestic equids with longer-chain EFAs may stem from the fact that these horses consume diets that contain more grains. Grains are typically higher in omega-6 EFAs [13], changing the ratio of intake of omega-3 to omega-6 fatty acids in domestic equine diets. As the ratio of omega-3:6 affects the conversion rate of ALA to EPA and DHA, there is interest in testing dietary sources of these longer-chain EFAs in equine diets.

There are many populations of domestic horses that consume grains as supplemental energy sources and would, therefore, potentially benefit from omega-3 supplementation; however, the data presented in this paper focus explicitly on late-gestation mares and neonatal foals. In other species, higher plasma DHA concentrations at birth are related to better visual acuity (human infants) [14] and reduced time to stand and suckle (lambs) [15]. Therefore, the goals of this project were to investigate the plasma fatty acid composition in foals prior to and after consumption of milk from mares supplemented with EFA sources beginning on day 310 of gestation. Our hypothesis was that foals born to mares provided with a fish oil supplement during late gestation would have greater plasma DHA concentrations than foals born to either control mares or mares supplemented with flaxseed-derived ALA.

## 2. Materials and Methods

Prior to the start of this study, Sam Houston State University’s Institutional Animal Care and Use Committee approved all procedures involving animals (protocol 20-02-25-2042-3-01).

### 2.1. Animals and Study Design

A controlled feeding trial consisting of fifteen late-gestation American Quarter Horse mares and their resultant foals was utilized to investigate the supplementation effects. For the feeding trial, the mares were assigned to 1 of 3 treatments: an unsupplemented control treatment (CON; n = 5 completed), a flaxseed (FLAX; n = 5 planned and n = 3 completed) omega-3 source, or a fish oil (FO; n = 5 completed) omega-3 source. The number of horses per treatment was determined based on a power analysis using data from previous research investigating plasma DHA concentrations in horses fed a fish oil supplement [16]. This study began in January 2019 (n = 11; 4 CON, 4 FO, and 3 FLAX) and ended in March 2020 (n = 2; 1 CON and 1 FO). A fourth flax horse could not be provided in 2019, and two years were required due to insufficient mares available during 2019 to achieve the sample size from our power analysis. An additional two mares for the FLAX treatment were planned for 2020, for a total plan of five mares per treatment; however, these mares did not deliver their foals prior to the beginning of the COVID-19 pandemic, which stopped research activity.

All mares were sourced from local breeders, and none were recipient carriers. The mares were transported from their home facilities on day 300 of gestation (average transportation time: 30–45 min) and maintained at Sam Houston State University for the duration of this project. The mares ranged in age from 5 to 22 y (with average ages of 16, 19, and 16 years for CON, FO, and FLAX, respectively) and in body weight from 461 to 573 kg. The mares began supplementation on gestational day 310, 30 days prior to their expected foaling date (EFD), and continued for 5 days after parturition. The expected foaling dates ranged from January to June. For this reason, the mares were assigned to the treatments in a sequential order of CON, FO, and FLAX to minimize the effects of the season, day length, and temperature.

All mares were housed individually in foaling stalls (3.66 m × 7.32 m) and allowed 1 h of daily turnout time in a covered arena, with no access to pasture. Five days after parturition, the mares and foals were returned to their respective owners, and supplementation ended. Continuing supplementation once the mares were returned to their owners was not conducive to the owners’ management practices of group housing mares and foals, as we would be unable to prevent mares from sharing buckets of feed with other mares. Follow-up samples, collected on day 30 of neonatal life, were collected at each animal’s home farm. The mares and foals could not be maintained at SHSU after 5 days of life due to a lack of sufficient turnout space that is needed for young foals to exercise. Routine vaccinations, deworming, and farrier schedules were maintained throughout this study. The same cutting of hay and brand of concentrate were used throughout the years of this study. Fresh sources of FO and FLAX were provided during the second year. Nutritional analyses were composited across the batches and years of this study.

### 2.2. Treatments and Diets for Feeding Trial

The basal ration for all 3 treatment groups included Coastal Bermudagrass hay, fed at 2% of BW, daily (Table 1). A grain-based concentrate (Calf Creep 14%, Producer’s Cooperative, Bryan, TX, USA; Table 1) was fed at 0.6% BW, daily. The hay and concentrate were split into two equal meals, fed at 0600 and 1800 h. Fresh water and salt were available ad libitum. The diet was formulated in accordance with the National Research Council’s recommendations for mares who are pregnant or lactating [17]. The foals also had access to their dams’ concentrate and supplement during the twice-daily feedings. Mares receiving treatment had their supplement top-dressed onto their individual basal rations. The FO treatment consisted of a fish oil source of DHA (DigestaWell Omega-3, Probiotech, Saint-Hyacinthe, QC, Canada), while the FLAX treatment consisted of a flax-based source of ALA (Smart & Simple Flax, SmartPak, Plymouth, MA, USA; Table 1). The supplementation of the mares began on day 310 of gestation and continued until day 5 post-partum. The FO and FLAX supplements were fed at a rate of 242.4 g and 190 g per day, respectively, with the amounts being chosen based on the labels’ claims of omega-3 fatty acid inclusion that would provide 40 g of omega-3 fatty acids per day. The supplementation rate of 40 g per day was chosen based on prior research that supplemented 38 g of omega-3 fatty acids daily [18,19]. Supplements were hand-mixed into the grain concentrate immediately prior to feeding, and no palatability problems were observed at the dosage fed. The supplements were stored at 4 °C until use.

### 2.3. Blood and Milk Collection

All blood samples were collected from the jugular vein into syringes and then transferred into heparinized, evacuated tubes (Vacutainer, BD, Franklin Lakes, NJ, USA). Blood samples were collected from pregnant mares on gestation day 309 (GEST309), prior to beginning supplementation on gestation day 310. Blood was also collected 12 h post-partum (PPD1) and on post-partum days 5 (PPD5) and 30 (PPD30). Blood was collected from the foals prior to suckling (NEOD0) and on post-partum days 5 (NEOD5) and 30 (NEOD30). Following the blood collection, the samples were centrifuged for 10 min at 1500× *g* and 4 °C, and the plasma was harvested and frozen at −80 °C until analysis. A blood sample was also collected from each foal at 12 h of life to measure the serum IgG concentration. These samples were placed in uncoated tubes (Vacutainer) and allowed to clot at room temperature for 2 h. The serum was harvested and stored in the same manner as the plasma.

Between 20 and 30 mL of milk was manually collected into 50 mL collection tubes from both teats of each mare on day 1 (PPD1) at approximately 12 h post-parturition. Milk collection was attempted at 5 (PPD5) and 30 (PPD30) days post-parturition but was predominantly unsuccessful, with inadequate volumes or a refusal to let down milk for hand collection. The milk was individually frozen at −80 °C until analysis.

### 2.4. Mare Body Condition Scores and Rump Fat Thickness

Body condition scores (BCSs) were appraised by the same researcher (Bedore) prior to starting supplementation (GEST309) at 12 h post-partum (PPD1) and 5 days (PPD5) post-partum. The scores were assigned using a 9-point scale [20], ranging from 1 (very poor) to 9 (extremely fat). The rump fat thickness of the mares was measured using B-mode ultrasonography prior to GEST309 and at 12 h (PPD1) and 5 (PPD5) days post-parturition. Measurements were performed at the rump, 5 cm lateral from the midline at the midpoint of the pelvic bone [21]. The region was scanned, and the position of maximal fat thickness was used as the measured site. The fat percentage was estimated using an equation, where % fat = 8.64 + (4.70 × cm rump fat) [21].

### 2.5. Foal Measurements and Time to Stand and Nurse

The foal weight was measured at 12 h (NEOD1) and 5 (NEOD5) days post-partum. The time to stand was measured by starting a timer when parturition was complete, and the stopping time was measured when the foal stood for a continuous 10 s. The time to nurse was measured by starting at the time that parturition was complete and ending when the foal had successfully latched onto the mare’s teat and was able to suckle. The foals were not assisted in either standing or nursing.

### 2.6. Serum IgG Analysis

Foal serum samples were collected at PPD1 and analyzed for their IgG contents using a commercially available ELISA (IMMUNO-TEK, ZeptoMetrix, Buffalo, NY, USA), according to the manufacturer’s specifications. The serum samples were diluted to 1:200,000 in an assay diluent for initial testing, and then 200 µL standards and 200 µL samples were pipetted into duplicate wells. The plates were covered and incubated for 30 min at 37 °C. The contents of each well were aspirated, and the wells were washed 4 times with 300 µL of a plate wash buffer. To each well, 100 µL of a conjugated antibody detector was added, and then the plates were covered and incubated for 30 min at 37 °C. The plates were once again washed 4 times, and then 100 µL of a substrate was added to each well. The plates were incubated for 30 min at room temperature, and 100 µL of a stop solution was added to each well. The optical density of each sample was measured at 450 nm and used to calculate the concentrations per the manufacturer’s instructions.

### 2.7. Milk Fat Extraction and Methylation

Lipids were extracted from milk fat cakes using 3:2 hexane–isopropanol (HIP; Fisher Scientific, Hampton, NH, USA), as previously described [22]. Briefly, the mare milk samples were thawed at room temperature and then weighed into 50 mL centrifuge tubes (Fisher Scientific) to a 40 g total weight. The samples were centrifuged at 5300× *g* for 30 min at 8 °C. The fat cakes were then removed, and 300–350 mg of the fat cakes was placed in a 16 × 100 screw-top test tube. Eighteen milliliters of HIP was added per g of fat cake, and then the samples were vortexed for 30 s. An amount of 12 milliliters of sodium sulfate solution (66.7 g of Na_2_SO_4_ per 1 L of H_2_O; Fisher Scientific) per g of fat cake was then added, and the mixture was once again vortexed for 30 s. The samples were then centrifuged at 1000× *g* for 5 min. The upper phase was transferred using a Pasteur pipette to a fresh 16 × 100 screw-top test tube containing 1 g of Na_2_SO_4_ and purged with nitrogen (Airgas, Radnor, PA, USA). After a 30 min incubation, the solvent was transferred to a fresh 16 × 100 screw-top test tube and loaded into a nitrogen evaporator with a dry bath set to 40 °C (Reacti-Vap, TS-18825, Thermo Fisher Scientific, Waltham, MA, USA). The solvent was then evaporated under a stream of nitrogen. Forty milligrams of the lipids was weighed out into a 16 × 100 screw-top test tube that was briefly purged with nitrogen gas and stored at −20 °C until methylation.

The extracted samples were warmed to room temperature, and 2 mL of hexane and 40 µL of methyl acetate (Fisher Scientific) were added. The samples were then vortexed for 30 s, 40 µL of a methylation reagent (0.4 mL of sodium methoxide and 1.75 mL of methanol; Fisher Scientific) was added, and then the samples were vortexed again for 2 min. After 8 min, 60 µL of a termination reagent (1 g of oxalic acid dissolved in 30 mL of diethyl ether; Fisher Scientific) was added, and then the samples were vortexed for 30 s. A small scoop of calcium chloride (Sigma-Aldrich, St. Louis, MO, USA) was added to each vial, and the vial was allowed to stand for 1 h. The samples were centrifuged at 2000× *g* for 5 min, and then the upper phase was transferred to a GC vial, which was purged with nitrogen gas and stored at −20 °C until analysis [23].

### 2.8. Plasma Fat Extraction and Methylation

The plasma samples were thawed at room temperature, and 3 mL of HIP was added to 2 mL of the plasma samples. Two milliliters of sodium sulfate solution was added, each sample was vortexed, and the upper phase was transferred to a fresh 16 × 100 screw-top tube containing 1 g of anhydrous sodium sulfate. Each sample was vortexed and incubated at room temperature for 40 min, after which the solution was transferred to a fresh 16 × 100 screw-top tube, and the solvent was evaporated under a stream of nitrogen. To this tube, 500 µL of hexane was added, as well as 40 µL of methyl acetate. Each sample was then vortexed, and 40 µL of a methylation reagent (0.4 mL of sodium methoxide and 1.75 mL of methanol) was added, and then each sample was vortexed again. The samples were allowed to react at room temperature for 24 h, and then 60 µL of a termination reagent was added. A few grains of calcium chloride were added to remove the water, and the upper phase was transferred to a GC vial for analysis [24].

### 2.9. Feed Lipid Extraction and Methylation

The hay, grain, and supplement samples were dried in a forced air oven (VWR, Radnor, PA, USA) at 60–70 °C for 4 h and then ground to pass through a 1 mm screen using a Wiley Mill (Thomas Scientific; Swedesboro, NJ, USA). The ground samples were weighed out to 500 mg and placed into 16 × 125 mm screw-top test tubes. The samples were directly methylated according to the method of Sukhija and Palmquist, using toluene in place of benzene [25].

### 2.10. Lipid Analysis

The fatty acid methyl esters from all samples (plasma, feeds, and milk) were analyzed by gas chromatography with a flame ionization detector using a DB-FastFAME 20 m × 0.18 mm i.d. column with a 0.2 μm film thickness (Agilent GC system 6890N, Agilent Technologies, Santa Clara, CA, USA) to determine the fatty acid compositions. The injector and detector temperatures were both 275 °C, with a flow rate of 0.64 mL/minute, which was splitless. The initial oven temperature was 150 °C and held for 1 min, and then increased at 4 °C/minute to 210 °C. The peaks corresponding to fatty acids were identified using pure standards (GLC-68D and -91, Nu-Chek Prep Inc., Elysian, MN, USA). To each sample, 100 µg of heptadecanoic acid was added prior to extraction as the internal standard (Nu-Chek Prep Inc.).

### 2.11. Statistical Analysis

All statistical analyses were performed using the MIXED procedure of SAS (v.9.4, SAS Inst., Inc., Cary, NC, USA). The nutrient intakes of the mares were analyzed using ANOVA for the effects of the treatments (CON, FO, and FLAX). The mares’ body weight, body fat%, and BCS were analyzed using ANOVA for the effects of the treatments on each day of this study. The foals’ body weight was analyzed using repeated-measures ANOVA for the effects of the day (NEOD1 and NEOD5) and treatments (CON, FO, and FLAX) and the day-by-treatment interactions. The foal plasma fatty acid analysis used repeated-measures ANOVA for the effects of the day (NEOD0, NEOD5, and NEOD30) and treatments (CON, FO, and FLAX) and the day-by-treatment interactions. The mare plasma fatty acid analysis used repeated-measures ANOVA for the effects of the day (GEST309, PPD1, PPD5, and PPD30) and treatments (CON, FO, and FLAX) and the day-by-treatment interactions. The mare milk fatty acid analysis used ANOVA for the effects of the treatments (CON, FO, and FLAX) on PPD1. Simple effect differences for time were detected by a Dunnett test to compare all time points with GEST309 (mares) or NEOD0 (foals), and simple effect differences for the treatments and the time-by-treatment interactions were determined using Tukey tests. The autoregressive covariance structure for the repeated analyses yielded the lowest AICC. The repeated term was the day of the study on the horse.

## 3. Results

### 3.1. Nutrient Ingestion

Mares assigned to CON, FO, and FLAX consumed similar daily quantities of dry matter (DM), DM as a percentage of body weight, and digestible energy (DE) (*p* > 0.2; Table 2). There was a main effect of the treatment on the ether extract (EE) intake (*p* < 0.001), whereby mares assigned to FO consumed more EE (428 ± 6) than both FLAX (321 ± 8; *p* < 0.001) and CON mares (281 ± 6 g; *p* < 0.001). Additionally, FLAX mares consumed more EE than CON mares (*p* = 0.008). The results for specific fatty acids are presented in Table 2. For omega-3 fatty acids, a main effect of the treatment was detected for ALA and EPA (*p* < 0.001), whereby FLAX horses consumed more of both of these fatty acids than CON or FO horses. There was no variance in the intake of DHA, with CON and FLAX mares consuming none and FO mares consuming 12.8 g per day, and due to the lack of variance, these data were not analyzed statistically. For omega-6 fatty acids, horses assigned to FO consumed more LA and 20:2 n-6 than either FLAX or CON horses (*p* < 0.001). There was no variance in the intake of 20:3 n-6, with CON and FO mares consuming none and FLAX mares consuming 0.1 g per day, and these data were not analyzed statistically. There was no variance in the intake of 20:4 n-6, with CON mares consuming none, FLAX mares consuming 0.05 g, and FO mares consuming 2.6 g per day, and due to the lack of variance, these data were not analyzed statistically. Mares assigned to the FO, FLAX, and CON diets consumed 36.86, 30.49, and 25.67 g of omega-3 fatty acids, respectively, on a daily basis.

### 3.2. Mare and Foal Morphometrics

There were no effects of the treatments on mare body weight, rump fat thickness, or BCS on any day of this study (*p* > 0.2; Appendix A). The mean foal weights were not affected by the time-versus-treatment interaction (*p* = 0.6; Table 3) but were different between treatments (*p* = 0.03).

### 3.3. Foal Time to Stand, Time to Nurse, Gestation Length, and Serum IgG Concentrations

No treatment differences in the time to stand were observed among the foals (*p* > 0.8). The mean times to stand were 32.6 min, 42.3 min, and 36.4 min for foals born to FO, FLAX, and CON mares, respectively. No treatment differences in the time to nurse were observed among the foals (*p* > 0.2). The mean times to nurse were 61.8 min, 98.3 min, and 83.8 min for foals born to FO, FLAX, and CON mares, respectively. No treatment differences in the length of gestation were observed among the mares (*p* > 0.4), with mean lengths of gestation of 337 days, 331 days, and 334 days for FO, FLAX, and CON mares, respectively. The mean serum IgG concentrations were 1502 mg/dL, 1158 mg/dL, and 1682 mg/dL for CON, FO, and FLAX foals, respectively, and no treatment differences were observed among the foals (*p* > 0.05).

### 3.4. Feeding Trial Foal Plasma Fatty Acids

The foal plasma was analyzed for the effects and interactions of the treatments and days (Table 4). For 16:0, 16:1 n-9, 18:1 n-9, and 18: 3 n-3, there were main effects of the day (*p* = 0.01, 0.002, <0.001, and 0.006, respectively). For 20:4 n-6 and EPA, there were effects for the treatment-by-day interaction (*p* = 0.06 and 0.001, respectively). For DHA, there was a main effect of the treatment (*p* = 0.045), whereby the mean percentage of DHA in plasma was higher in foals born to FO mares (0.8 ± 0.1%) than foals born to CON mares (0.3 ± 0.1%; *p* = 0.039) but was not different from that of foals born to FLAX mares (0.4 ± 0.1%; *p* = 0.2). Foals born to FLAX and CON mares had similar mean plasma percentages of DHA (*p* = 0.6).

### 3.5. Mare Plasma and Milk Fatty Acids

The analysis of the mare plasma for FAs resulted in no significant differences observed for the treatments by day for any fatty acid (*p* > 0.1; Table 5). There was a main effect of the day for 16:0 (*p* < 0.001), a main effect of the treatment for 20:2 n-6 (*p* = 0.03), a tendency for a main effect of the day (*p* > 0.07) for 20:2 n6, and a main effect of the day for 20:3 n-6 (*p* = 0.005). The mare milk yielded no observed differences in the effects of the treatment (*p* > 0.3; Table 6).

## 4. Discussion

The finding of increased plasma proportions of DHA (22:6 n-3) in foals born to FO-supplemented mares is consistent with other studies indicating that the dietary supplementation of marine-derived omega-3 fatty acids (fish oil and algae) to mares in late gestation and early lactation results in increased plasma DHA levels in their foals [26,27,28]. This could be due to the FO mares consuming more DHA than the FLAX or CON mares and suggests that ingested DHA is preferentially transferred from mares to foals during gestation, as the mares did not experience increased plasma or milk DHA contents. Fetal transfer is observed in humans, with fetal erythrocytes playing a role in the uptake of DHA from maternal blood [29], and cord blood having higher DHA concentrations than maternal blood [30]. These human findings suggest some degree of preferential transfer of specific fatty acids, including DHA [31,32]. In mares, Stammers et al. [33] observed a positive correlation between maternal and umbilical vein plasma free fatty acid concentrations, which included substantial amounts of essential fatty acids and their longer-chain derivatives, suggesting that the equine placenta is permeable to long-chain fatty acids. Cumulatively, these observations support the hypothesis that there is placental EFA transfer to the equine fetus when mares are supplemented during late gestation. Interestingly, there was no difference in the EPA and DHA concentrations at birth between foals born to FO- or FLAX-treated mares, which could be due to the low amount of power in the FLAX treatment. The original planning included the sampling of additional horses; however, this was prevented by the COVID-19 pandemic. Secondly, although our goal was to feed an additional 40 g of omega-3 fatty acids through the FO and FLAX supplements, we only achieved an additional 11.19 and 4.82 g of omega-3 fatty acids per day, respectively. This lower-than-desired level of supplementation could have impacted our results.

Mammalian neonates can meet their DHA requirement, in part, through the elongation of the 18-carbon precursor, ALA [34,35], which is abundant in mare milk and human breast milk. In humans, the remaining requirement for DHA is met through either breast milk or infant-specific formulas, with these dietary sources being necessary because the elongase and desaturase enzymes that produce DHA are insufficiently expressed in neonates to meet their high requirement [36]. The proportions of mare milk DHA reported herein are similar to reports in human breast milk [37], with DHA being present in low amounts (0.14 to 0.37%) [38]. Interestingly, Inuit mothers, with an exceptionally high dietary intake of DHA, do provide greater amounts of DHA in their breast milk, with proportions reaching 1.2% [36]. Our FO mares did not experience an increased DHA content in their milk, which was likely due to the lower-than-expected dietary inclusion rate. The patterns of the DHA proportions in our foal plasma across days suggest that an equine fetus, similar to a human fetus, is provided with substantial DHA during gestation, with a post-birth shift to a greater reliance on the elongation of ALA to meet their DHA requirement due to a lower supply in their diet. Keeping in mind that our FLAX and FO mares ceased supplementation on day 5 of life, it is interesting that the pattern of the plasma DHA content in CON foals was similar to that observed in the other treatment groups. This decline in foal plasma concentrations was similarly reported by Kouba et al. [28]. Simultaneously, there is a greater reliance on ALA desaturation and elongation to supply DHA; however, ALA is supplied in a greater quantity in milk and feed than DHA. Therefore, it is possible that these dietary sources account for ALA’s increasing concentration in plasma post-birth despite its concurrent usage to supply DHA.

The most abundant fatty acids in the mares’ milk were LA, 18:1 n-9, 16:0, and 14:0. These patterns are similar to the fatty acid profiles of human breast milk, which also provides high percentages of 18:1 n-9, LA, and 16:0 and low percentages of fatty acids of greater than 18 carbons. In other species, including pigs and humans, 18:1 n-9 is a critical fatty acid for brain myelination and is positively associated with learning in the neonatal period [39,40]. Furthermore, 16:0 in the middle position of a triglyceride increases fatty acid absorption in the gastrointestinal tract [41], indicating its importance and a rationale for its high levels in milk. The current study observed minimal levels of DHA in the mares, which is similar to the findings reported by Duvaux-Ponter et al. [42], who reported not observing any DHA in mare milk when supplementing with either rapeseed or linseed. Additionally, Pikul and Wojtowski [43] were not able to detect any EPA or DHA when sampling mare milk for its fatty acid composition. Similarly, in humans, women supplemented with flaxseed oil did not have increased DHA levels in their breast milk [44]. Paired with the results of the foal plasma, this suggests the most effective avenue for supplying a foal with DHA is in utero through maternal supplementation during gestation.

While both supplements provided ALA, FLAX mares consumed more. Despite this, there was no treatment effect on foal ALA percentages at birth. It is possible this was due to the preferential placental transfer of longer-chain fatty acids or the elongation of ALA to EPA and DHA. Intriguingly, the pattern of ALA in neonatal plasma over time opposed that of DHA, with a steady increase throughout the 30 d post-parturition sample collection. Similarly, Kouba et al. [28] observed that foal plasma ALA content increased significantly after birth and stayed elevated through 21 days of life. It is possible that this was due, in part, to foal feed consumption, as foals begin consuming pasture and hay within one week of life [45], and these feeds are excellent sources of ALA.

We also observed that the omega-6 fatty acid LA increased from birth. This is similar to the results of Duvaux-Ponter et al. [42], who observed that LA levels in foal plasma increased after suckling. As observed in our CON mares and foals, LA is the most abundant fatty acid in foal plasma and mare milk. Similarly, Kouba et al. [28] also observed that LA was the most abundant fatty acid in foal plasma and mare milk across treatments and time. Linoleic acid can be elongated and desaturated to longer-chain omega-6 fatty acids, and these fatty acids can be converted to prostaglandins and leukotrienes. These compounds are essential components of inflammation that initiate the repair of damaged and infected tissues [46]. Linoleic acid is also an essential component of the epidermis, necessary for maintaining membrane fluidity and waterproofing, as noted in both humans and horses [47,48]. Furthermore, linoleic acid serves as an energy source, with human infants deriving at least 2% of their energy from linoleic acid [49]. These roles indicate a high need for linoleic acid in growing foals and a rationale for the increased plasma concentrations following suckling.

While previous research has observed increased DHA levels in the plasma of mares supplemented with fish oil [27], we did not observe any differences in the plasma fatty acid levels of either FO or FLAX compared with CON mares. The reason for this inconsistency may be due to differences in the experimental design, with the mares in the present study only consuming an additional 11.19 and 4.82 g of omega-3 fatty acids per day, whereas previous studies offered higher quantities. Additionally, it was observed that mare plasma EPA levels were mostly not detectable throughout this study, regardless of the treatment. Adkin [27] also reported this finding when supplementing mares with an algal supplement. In contrast, Kouba et al. [28] observed that when mares were supplemented with DHA and EPA (10.43 g and 8.84 g daily, respectively) for 60 days prior to parturition, they expressed higher levels of EPA and DHA in both plasma and milk. Furthermore, the current study was also inconsistent with findings in non-pregnant animals that had been supplemented with fish oil or algae. For example, King et al. [50] observed significantly higher EPA and DHA plasma concentrations in mares fed 681 g/day of a marine-derived supplement designed to supply 10, 20, or 40 g/day of EPA and DHA depending on the treatment. Similarly, Hess et al. [26] observed elevated EPA and DHA concentrations in mares fed a marine-derived omega-3 fatty acid supplement providing 2 g of ALA, 7.6 g of EPA, 26.6 g of DHA, and 1.7 g of 22:5 n-3 acid (DPA) daily. That specific study did not observe any detectable levels of EPA or DHA in mares supplemented with a flax supplement. It is possible that a specific inclusion rate is necessary to see changes in the plasma composition, and that our DHA feeding rates did not meet this break point. Lastly, FLAX mares experienced greater mean proportions of 20:2 n6, an omega-6 fatty acid that is formed during the elongation of LA to form arachidonic acid [51], which could suggest that greater substrate availability increased elongation rates. Given the low number of mares used for the FLAX treatment, this finding should be investigated further.

A major limitation of this study was the small sample size of the FLAX mares and foals. This was due to the onset of the COVID-19 pandemic, which forced us to end our sample collection early. This small sample size could have contributed to the differences in foal weights. Although not statistically significant, the FLAX mares also weighed less than mares in the other treatment groups, and with greater power, these differences might not have been observed. Whether flaxseed oil influences foal weights should be investigated with large sample sizes, as the mare diets were kept isocaloric.

## 5. Conclusions

The dietary supplementation of mares with fish oil during late gestation alters the fatty acid composition of the foal plasma at birth, resulting in higher proportions of 22:6 n-3 compared with foals from unsupplemented mares. Furthermore, major shifts in plasma lipid concentrations occur after birth and the initiation of suckling, with linoleic acid becoming the predominant fatty acid within 5 d of life. Mares supplemented with FLAX may have increased proportions of 20:3 n6, which could indicate higher rates of LA elongation to arachidonic acid. Lastly, researchers should aim to test supplements prior to beginning research trials in order to ascertain nutrient levels. This can also allow the testing of palatability if the volumes of supplements need to be adjusted to meet targeted nutrient consumption levels.

## Figures and Tables

**Table 1 animals-15-01612-t001:** Nutrient composition of dietary components fed to late-gestation mares for a minimum of 30 days.

Nutrient (DM Basis) ^1^	Hay ^2^	Concentrate ^3^	FO ^4^	FLAX ^5^
DE, MCal/kg	0.82	1.43	2.85	2.16
Crude protein	9.5	22.8	0.5	24.1
ADF	39.1	11.0	20.6	17.1
NDF	72.2	25.2	36.8	24.2
EE	2.0	3.0	72.1	32.3
Ash	7.35	6.54	7.73	4.12
Fatty acid percentage, %				
14:0	0	0.14	1.76	0
14:1 n9	0	0	0	0
16:0	27.79	16.10	1.62	0.52
16:1 n9	19.36	0.18	22.48	5.34
18:0	0	1.81	1.13	0
18:1 n9	10.32	25.65	0	0
18:2 n6	7.66	48.70	38.19	29.06
18:3 n3	11.90	2.79	0	11.22
20:0	0	0.38	2.35	52.00
20:1 n9	2.44	0	0	0
20:2 n6	0	0.05	4.80	0.30
20:3 n6	0	0	0	0.05
20:4 n6	0	0	1.55	0.08
20:5 n3	0	0.08	0	0.05
22:0	4.31	0.27	13.17	0.05
22:1 n9	2.81	0	0	0.31
22:6 n3	0	0	7.64	0

^1^ DE = digestible energy; ADF = acid detergent fiber; NDF = neutral detergent fiber; EE = ether extract; ^2^ Coastal Bermudagrass (Cynodon dacytlon); ^3^ Calf Creep 14%, Producer’s Cooperative, Bryan, TX, USA; ^4^ DigestaWell Omega-3, Probiotech International, Saint-Hyacinthe, QC, Canada; ^5^ Smart and Simple Flax, SmartPak, Plymouth, MA, USA.

**Table 2 animals-15-01612-t002:** Daily average nutritional intakes of mares consuming Coastal Bermudagrass and grain-based concentrate that were supplemented daily with 242.4 g of fish oil (FO) or 190 g of flax (FLAX) daily for 30 days prior to expected parturition or remaining non-supplemented controls (CON) ^1,2^.

Nutrient ^3^	CON	FO ^4^	FLAX ^5^	*p*-Value
DM, kg	12.6 ± 0.3	12.0 ± 0.3	12.0 ± 0.4	0.2
DM, % of body weight	2.3 ± 0.01	2.3 ± 0.01	2.3 ± 0.02	0.4
Estimated DE, Mcal	23.6 ± 0.6	22.2 ± 0.6	22.2 ± 0.8	0.2
Provided DE, Mcal	26.6 ± 0.6	26.1 ± 0.6	25.8 ± 0.8	0.7
CP, g	1577 ± 34 ^i^	1457 ± 34 ^j^	1523 ± 44 ^ij^	0.094
ADF, kg	4.1 ± 0.1	3.9 ± 0.1	3.9 ± 0.1	0.2
NDF, kg	7.8 ± 0.2	7.4 ± 0.2	7.3 ± 0.2	0.2
EE, g	281 ± 6 ^a^	428 ± 6 ^c^	321 ± 8 ^b^	<0.001
14:0	0.1 ± 0.003 ^a^	3.1 ± 0.003 ^b^	0.1 ± 0.003 ^a^	<0.001
16:0	68.0 ± 1.6	66.1 ± 1.6	64.7 ± 2.1	0.44
18:0	1.5 ± 0.03 ^a^	3.3 ± 0.03 ^b^	1.5 ± 0.04 ^a^	<0.001
20:0	0.3 ± 0.01 ^a^	4.2 ± 0.01 ^b^	87.6 ± 0.01 ^c^	<0.001
22:0	8.4 ± 0.2 ^a^	30.0 ± 0.2 ^b^	8.0 ± 0.3 ^a^	<0.001
16:1 n-9	85.6 ± 1.8 ^a^	114.8 ± 1.8 ^b^	89.5 ± 2.4 ^a^	<0.001
18:1 n-9	42.1 ± 0.9 ^i^	38.7 ± 0.9 ^j^	39.5 ± 1.2 ^ij^	0.058
20:1 n-9	4.8 ± 0.1	4.5 ± 0.2	4.5 ± 0.2	0.2
22:1 n-9	5.5 ± 0.1	5.2 ± 0.1	5.7 ± 0.2	0.11
18:3 n-3	25.6 ± 0.6 ^a^	24.0 ± 0.6 ^a^	30.4 ± 0.8 ^b^	0.001
20:5 n-3	0.07 ± 0.001 ^b^	0.06 ± 0.001 ^a^	0.09 ± 0.001 ^c^	<0.001
22:6 n-3	0.0	12.8	0.0	---
18:2 n-6	56.5 ± 1.2 ^a^	115.6 ± 1.2 ^c^	69.8 ± 1.5 ^b^	<0.001
20:2 n-6	0.04 ± 0.001 ^a^	8.1 ± 0.001 ^c^	0.5 ± 0.001 ^b^	<0.001
20:3 n-6	0.0	0.0	0.1	---
20:4 n-6	0.0	2.6	0.05	---
Others	0.3 ± 0.01	7.4 ± 0.01	0.3 ± 0.01	<0.001

^1^ Diet consisted of Coastal Bermudagrass (Cynodon dactylon; 2% of body weight) and grain-based concentrate (0.6% of body weight; Calf Creep 14%, Producer’s Cooperative, Bryan, TX, USA); ^2^ standard error of the mean; ^3^ DE = digestible energy; CP = crude protein; ADF = acid detergent fiber; NDF = neutral detergent fiber; EE = ether extract; ^4^ diet included an additional 242.4 g of DigestaWell Omega-3, Probiotech International, Saint-Hyacinthe, QC, Canada; ^5^ diet included an additional 190 g of Smart and Simple Flax, SmartPak, Plymouth, MA, USA; ^abc^ within-row means, with unlike superscripts differing at *p* < 0.05; ^ij^ within-row means, with unlike superscripts differing at *p* < 0.1.

**Table 3 animals-15-01612-t003:** Mean body weights (kg) of foals born to unsupplemented mares (CON) and mares supplemented daily with 242.4 g of fish oil (FO) or 190 g of flax (FLAX), beginning on day 310 of gestation.

Treatment (TRT)	TRT Mean	Day of Life ^3^		SEM ^4^	*p*-Value		
		NEOD1	NEOD5		TRT	Day	TRT × Day
CON	50.6 ^a^ *	47.4	53.8	2.7			
FO ^1^	44.0 ^b^ *	41.5	46.5	2.7	0.005	0.013	0.884
FLAX ^2^	39.4 ^bc^	35.5	43.3	3.1			

^1^ DigestaWell Omega-3, Probiotech International, Saint-Hyacinthe, QC, Canada; ^2^ Smart and Simple Flax, SmartPak, Plymouth, MA, USA; ^3^ foals were evaluated on days 1 (NEOD1) and 5 (NEOD5) of foal life; ^4^ standard error of the mean; ^abc^ means with unlike superscripts differ at *p* < 0.05; * means with unlike superscripts tend to differ at *p* < 0.1.

**Table 4 animals-15-01612-t004:** Mean ± SEM plasma fatty acid proportions of total plasma lipids in foals born to mares supplemented with either 242.4 g of fish oil (FO ^1^) or 190 g of flax (FLAX ^2^) beginning on day 310 of gestation or in unsupplemented controls (CON) when collected prior to suckling (NEOD0) and at 5 (NEOD5) and 30 (NEOD30) days of life ^3^.

Treatment (TRT)	Day (D) Post-Parturition				*p*-Value		
Fatty Acid, %	NEOD0	NEOD5	NEOD30	SEM	TRT	D	TRT × D
16:0					0.9	0.01	0.3
D mean ^4^	27.4	22.9 *	19.2 **	1.9			
CON	30.8	22.0	15.6	3.0			
FO	26.1	23.0	21.4	2.8			
FLAX	25.3	23.8	20.7	3.7			
16:1 n9					0.4	0.003	0.1
D mean	2.3	1.3 **	1.5	1.3			
CON	5.1	3.2	3.6	2.0			
FO	1.2	0.7	0.6	1.0			
FLAX	0.5	0.1	0.4	2.6			
18:0					0.8	0.1	0.4
D mean	12.4	8.6	9.9	2.3			
CON	10.8	7.6	10.6	3.8			
FO	13.0	12.4	10.5	3.5			
FLAX	13.4	5.6	8.9	4.6			
18:1 n9					0.3	0.002	0.3
D mean	23.4	11.2 **	8.7 **	2.8			
CON	20.8	16.7	15.7	4.7			
FO	25.7	7.1	3.6	4.2			
FLAX	23.6	9.8	6.8	5.4			
18:2 n6					0.6	<0.001	0.3
D mean	23.1	39.5 **	46.0 ***	4.0			
CON	21.4	40.6	48.5	6.7			
FO	20.7	45.1	53.1	6.2			
FLAX	27.1	32.8	36.4	8.0			
18:3 n3					0.1	0.006	0.09
D mean	0.4	6.8 **	7.9 **	2.2			
CON	0.4	1.8	2.6	3.6			
FO	0.2	3.2	4.2	3.4			
FLAX	0.2	1.0	1.0	0.2			
20:4 n6					0.2	0.1	0.041
D mean	0.00	0.10	0.04	0.04			
CON	0.00 ^a^	0.00 ^a^	0.04 ^a^	0.06			
FO	0.00 ^a^	0.00 ^a^	0.07 ^b^	0.06			
FLAX	0.00 ^a^	0.30 ^b^	0.00 ^a^	0.07			
20:5 n3					0.1	0.01	0.001
D mean	0.15	0.27	0.01	0.08			
CON	0.00 ^a^	0.02 ^a^	0.02 ^a^	0.13			
FO	0.26 ^a^	0.79 ^b^	0.00 ^a^	0.13			
FLAX	0.18 ^ab^	0.00 ^a^	0.00 ^a^	0.15			
22:6 n3					0.045	<0.001	0.097
D mean	1.2	0.3 ***	0.03 ***	0.1			
CON ^a^	0.6	0.2	0.01	0.2			
FO ^b^	1.9	0.6	0.1	0.2			
FLAX ^ab^	1.1	0.2	0.0	0.3			

^1^ DigestaWell Omega-3, Probiotech International, Saint-Hyacinthe, QC, Canada; ^2^ Smart and Simple Flax, SmartPak, Plymouth, MA, USA; ^3^ standard error of the mean; ^4^ D mean indicates the mean of all three treatments on a given day of this study; ^ab^ means with unlike superscripts differed at *p* < 0.05; day means differ from day 0 at *p* < 0.05 * *p* < 0.01 ** or *p* < 0.001 ***.

**Table 5 animals-15-01612-t005:** Mean plasma fatty acid proportions of total plasma fatty acids from mares supplemented with either 242.4 g of fish oil (FO ^1^) or 190 g of flax (FLAX ^2^) beginning on day 310 of gestation or in unsupplemented controls (CON).

Treatment (TRT)	Day (D) Post-Parturition					*p*-Value		
Fatty Acid, %	GEST309	PPD1	PPD5	PPD30	SEM ^3^	TRT	D	TRT × D
14:0						0.9	0.082	0.3
D mean	0.9	0.7	0.1	0.3	0.3			
CON	1.6	0.5	0.0	0.3	0.4			
FO	0.5	0.8	0.4	0.3	0.4			
FLAX	0.6	0.8	0.3	0.3	0.5			
16:0						0.3	<0.001	0.4
D mean	17.1 ^a^	21.1 ^b^	19.5 ^bc^	18.4 ^ac^	1.3			
CON	15.9	18.1	15.5	14.6	1.9			
FO	16.5	20.2	18.9	19.2	1.9			
FLAX	18.8	25.1	24.2	21.5	2.5			
16:1 n9						0.5	0.7	0.6
D mean	1.0	1.1	1.3	1.4	1.4			
CON	2.9	3.0	3.9	4.2	2.3			
FO	0.1	0.1	0.1	0	2.3			
FLAX	0	0.2	0	0	2.9			
18:0						0.5	0.2	0.7
D mean	17.7	13.5	16.1	15.5	2.3			
CON	17.3	8.9	13.8	15.2	3.5			
FO	21.0	16.4	18.2	16.7	3.5			
FLAX	14.7	15.2	16.3	14.5	4.6			
18:1 n9						0.3	0.2	0.8
D mean	4.6	6.0	5.0	5.0	1.9			
CON	6.8	9.80	9.3	9.7	3.0			
FO	4.7	4.9	3.6	3.5	3.0			
FLAX	2.2	3.9	1.9	1.9	3.9			
18:2 n6						0.5	0.8	0.1
D mean	51.8	53.6	54.2	54.4	3.1			
CON	43.3	53.5	52.0	49.2	4.8			
FO	53.1	54.4	55.2	57.1	4.8			
FLAX	58.9	52.8	55.5	57.0	6.3			
18:3 n3						0.4	0.5	0.8
D mean	2.9	2.0	1.0	1.7	0.9			
CON	5.2	2.8	0.7	2.4	1.4			
FO	1.7	1.2	0.8	1.8	1.4			
FLAX	1.7	1.9	1.6	0.9	1.8			
20:2 n6						0.03	0.07	0.4
D mean	0.2	0.4	0.3	0.6	0.1			
CON ^a^	0.1	0.4	0.1	0.2	0.2			
FO ^a^	0.1	0.1	0.2	0.2	0.2			
FLAX ^b^	0.3	0.7	0.6	1.3	0.3			
20:3 n6						0.2	0.005	0.4
D mean	1.4 ^a^	1.2 ^a^	2.0 ^b^	1.1 ^a^	0.2			
CON	1.5	0.8	1.6	0.7	0.3			
FO	1.3	1.4	2.7	1.2	0.3			
FLAX	1.3	1.2	1.9	1.5	0.4			
22:6 n3						0.2	0.5	0.2
D mean	0.4	0.1	0.5	0.9	0.4			
CON	1.3	0.3	1.5	0.3	0.7			
FO	0	0.04	0.2	0	0.7			
FLAX	0	0	0	2.5	0.8			

^1^ DigestaWell Omega-3, Probiotech International, Saint-Hyacinthe, QC, Canada; ^2^ Smart and Simple Flax, SmartPak, Plymouth, MA, USA; ^3^ standard error of the mean; ^abc^ means with unlike superscripts differ at *p* < 0.05.

**Table 6 animals-15-01612-t006:** Mean (±SEM ^1^) milk fatty acid proportions of total milk fat from mares supplemented with either 242.4 g of fish oil (FO ^2^) or 190 g of flax (FLAX ^3^) beginning on day 310 of gestation or in unsupplemented controls (CON) when determined one day post-parturition.

	Treatment (TRT)			
Fatty Acid, %	CON	FO	FLAX	*p*-Value
14:0	5.8 ± 1.4	6.7 ± 1.2	5.5 ± 1.6	0.8
14:1 n9	1.8 ± 2.9	3.8 ± 2.6	0.2 ± 3.4	0.7
16:0	21.2 ± 2.8	22.3 ± 2.5	24.4 ± 3.3	0.7
16:1 n9	0.05 ± 0.03	0.05 ± 0.03	0.08 ± 0.04	0.7
18:1 n9	29.0 ± 4.7	23.9 ± 3.7	22.6 ± 4.7	0.5
18:2 n6	31.2 ± 3.5	32.0 ± 2.7	31.3 ± 3.5	0.9
18:3 n3	3.3 ± 2.1	2.3 ± 1.9	3.8 ± 2.4	0.8
20:0	0.3 ± 0.2	0.5 ± 0.2	0.7 ± 0.2	0.3
20:2 n6	1.1 ± 0.4	0.8 ± 0.3	0.9 ± 0.4	0.8
20:3 n6	0.00 ± 0.03	0.05 ± 0.02	0.03 ± 0.03	0.4
20:4 n6	0.7 ± 0.4	0.1 ± 0.3	0.4 ± 0.4	0.4
20:5 n3	0.04 ± 0.07	0.10 ± 0.06	0.00 ± 0.08	0.5
22:6 n3	0.5 ± 0.6	1.2 ± 0.5	0.5 ± 0.7	0.6

^1^ Standard error of the mean; ^2^ DigestaWell Omega-3, Probiotech International, Saint-Hyacinthe, QC, Canada; ^3^ Smart and Simple Flax, SmartPak, Plymouth, Massachusetts.

## Data Availability

The data will be made available upon request. Please send inquiries to jksuagee@vt.edu.

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
