# Peer review of "Mare Milk and Foal Plasma Fatty Acid Composition in Foals Born to Mares Fed Either Flax or Fish Oil During Late Gestation"

_animals, 2025, doi:10.3390/ani15111612_

Round 1
Reviewer 1 Report
Comments and Suggestions for Authors
Overall, an interesting paper to read, and well written. Just a few minor edits/comments, as listed below.
Line specific comments:
Line 58-59: is there a parenthesis missing in front of DHA? Please double check
Line 367: just to clarify, the data presented in lines 367-378 (Section 3.4) is from different mare/foal pairs as compared to the data in sections 3.1, 3.2, 3.3, 3.5, and 3.6? It is a little jarring to me as a reader to be shown that data in the middle of results from the other part of the project. I had to double check myself to make sure what I read before that was different horses. I would find it easier to read if the survey data was presented as 3.1, then present the results from the controlled trial.
Line 527: add a comma after LA
Author Response
Overall, an interesting paper to read, and well written. Just a few minor edits/comments, as listed below.
Response- thank you for the helpful feedback!
Line specific comments:
Line 58-59: is there a parenthesis missing in front of DHA? Please double check
Response- thank you for catching this, we have added in the parenthesis.
Line 367: just to clarify, the data presented in lines 367-378 (Section 3.4) is from different mare/foal pairs as compared to the data in sections 3.1, 3.2, 3.3, 3.5, and 3.6? It is a little jarring to me as a reader to be shown that data in the middle of results from the other part of the project. I had to double check myself to make sure what I read before that was different horses. I would find it easier to read if the survey data was presented as 3.1, then present the results from the controlled trial.
Response- as requested by the other reviewers, these data have been removed from the currently presented data.
Line 527: add a comma after LA
Response- thank you also for catching this grammatical error. We have corrected this.
Reviewer 2 Report
Comments and Suggestions for Authors
Major comments:
This study provides interesting data. However, the major limitations are the small study number, and the disappointing rates of supplementation of omega-3s which means that the relevance of the study is slightly questionable. Greater analysis of the supplements given in the pilot study would have improved the validity of the study?
Greater discussion on the limitations of the study is needed. The study numbers are very small and especially with the reduced numbers the study was unlikely to reach statistical significance in many areas. This is not adequately discussed.
Specific comments:
Line 37: descriptive terms for fatty acids may be better in the abstract to make more accessible for general reader
Line 111: this preliminary study obviously provides important data but maybe better presented separately? It adds some confusion to the rest of the data and is not mentioned in the title or abstract.
Line 129: you detail that the horses were assigned in a specific order but this should mean that there were 4 in the flax group? Please add a comment about this.
Line 145: could you comment on why supplementation ended? You detail why the horses could not remain on site and hence diet could be less well controlled, but testing at day 30 seems a little pointless without ongoing supplementation?
Line 171: were there any palatability problems? There is no mention of any discussion about ensuring full ration was fed.
Table 1: there is so much data presented in this study, perhaps this table could be provided as supplementary information?
Line 309: there is some repetition between the information presented in the table and the text which could be reduced.
Line 341: the study time period was very short to see major changes in weight or body fat? And supplementation so late in gestation is surely unlikely to have major effects on foal weight? These tables could be added to supplementary information.
Line 347: I have concerns about the relevance of these weight differences, especially which such small numbers. This needs more discussion.
Table 5: see comments above
Line 378: some of the repetition between text and table could be reduced
Line 481: what is the relevance of this to equines? Nervous tissue development is much more complete in foals compared to humans, what is the relevance of late term supplementation? Surely supplementation earlier in gestation is more likely to be beneficial?
Line 498: these changes are interesting, but I wonder about the significance of the 30 day results when mares were no longer being supplemented.
Author Response
This study provides interesting data. However, the major limitations are the small study number, and the disappointing rates of supplementation of omega-3s which means that the relevance of the study is slightly questionable. Greater analysis of the supplements given in the pilot study would have improved the validity of the study?
Response-Although sample size is small; we did meet our power analysis for both control and FO supplemented mares. We unfortunately had to finish the study early before we could meet the targeted sample size for FLAX mares. We were also disappointed by these unfortunate events.
A pilot study prior to testing supplements is a great idea, and we have added a statement to this effect to the conclusion. While we had no issues with palatability at the administered level, it is possible that feeding at a higher rate would reveal palatability issues.
Greater discussion on the limitations of the study is needed. The study numbers are very small and especially with the reduced numbers the study was unlikely to reach statistical significance in many areas. This is not adequately discussed.
Response- Thank you for pointing out this omission. We have added a final paragraph on the limitations of the study.
Specific comments:
Line 37: descriptive terms for fatty acids may be better in the abstract to make more accessible for general reader
Response- We agree, descriptive terms for fatty acids have been added to the abstract.
Line 111: this preliminary study obviously provides important data but maybe better presented separately? It adds some confusion to the rest of the data and is not mentioned in the title or abstract.
Response- Data from the preliminary study was removed per suggested from all reviewers
Line 129: you detail that the horses were assigned in a specific order but this should mean that there were 4 in the flax group? Please add a comment about this.
Response- Unfortunately, the breeder that normally provided mares for our teaching and research needs experienced difficulties that year and did not have as many bred mares as normal. As FLAX was the third treatment, we ran into the end of the foaling season and no more mares were available for the study, hence why only 3 were used for FLAX during year 1. A statement was added to this effect.
Line 145: could you comment on why supplementation ended? You detail why the horses could not remain on site and hence diet could be less well controlled, but testing at day 30 seems a little pointless without ongoing supplementation?
Response- Thank you for this feedback. Continuing the supplementation after day 5 was not considered possible by the owners as the mares went out on turnout with other mares and foals and continuing individual feeding would have required additional management that owners were unwilling to provide. While the grain and hay could be kept the same, it was keeping them confined at one bucket that was prohibitive.
The collection of samples on day 30 was more objective driven than hypothesis driven, in that we desired to know the direction of change for fatty acids supplementation ended. These data can be removed if desired by the reviewer. Further description has been added to the manuscript.
Line 171: were there any palatability problems? There is no mention of any discussion about ensuring full ration was fed.
Response- we did not observe any palatability problems with these products. A statement was added to this effect.
Table 1: there is so much data presented in this study, perhaps this table could be provided as supplementary information?
Response- we are happy to move some of the data to supplementary information. We do feel that the nutrient composition of the diet is customarily provided with the manuscript. Therefore, we wish to retain table 1 in the main submission.
Line 309: there is some repetition between the information presented in the table and the text which could be reduced.
Response- repetition has been reduced as requested.
Line 341: the study time period was very short to see major changes in weight or body fat? And supplementation so late in gestation is surely unlikely to have major effects on foal weight? These tables could be added to supplementary information.
Response- As requested, this table has been moved to supplemental data.
Line 347: I have concerns about the relevance of these weight differences, especially which such small numbers. This needs more discussion.
Response: discussion has been added to end of discussion section.
Table 5: see comments above
Response: Table 5 comments are in relation to the above point.
Line 378: some of the repetition between text and table could be reduced
Response- we have worked to reduce repetition.
Line 481: what is the relevance of this to equines? Nervous tissue development is much more complete in foals compared to humans, what is the relevance of late term supplementation? Surely supplementation earlier in gestation is more likely to be beneficial?
Response- We have removed this first sentence.
Line 498: these changes are interesting, but I wonder about the significance of the 30 day results when mares were no longer being supplemented.
Response- Thank you for this feedback. We agree that this is a limitation of the study, however, since the pattern observed holds up in the control horses, we have added a statement to the effect. There was a tendency for an interaction of time by treatment and we would be willing to reanalyze the data looking a tukey tests instead of main effects if that is what the reviewer suggests.
Reviewer 3 Report
Comments and Suggestions for Authors
ARTICLE
“Plasma fatty acid composition of foals born to mares fed either flax or fish oil during late gestation”
BRIEF SUMMARY AND GENERAL COMMENTS
This study compared the influence of supplementing flaxseed and fish oil at the end of gestation and peripartum on the plasma fatty acid profile of mares and foals and on mares’ milk composition. The current study provides a knowledge base for promoting the health of newborn foals through maternal nutrition.
Overall, the study was well designed, analysed, and written, and presented a very interesting discussion. Some attention is necessary to meet the journal’s standards in terms of tabulations and spacing between subtopics. Some statements were provided to enhance the quality of this study.
The secondary objective lacks background information regarding the nutritional description and composition of diets, comprising a group of animals allocated to farms that apparently did not relate at all with the main data, making it difficult to justify the inclusion of these data, despite being potentially interesting. I suggest removing them from the article and correcting the related sections. It could perhaps be presented at some congress and indirectly cited throughout the discussion.
I take this opportunity to present some suggestions, which I believe would be of great value in the evaluation of foal health and could be considered as suggestions for further publications. It would be interesting to know whether there was any treatment effect on morbidity and/or mortality during the first week, first month, and until weaning. Would it be possible to retrieve this data for another study?
SPECIFIC COMMENTS
Title
It should also include maternal parameters, as it was part of the study objective.
Simple Summary and Abstract
Corrections must be addressed by focusing on the objectives presented and the conclusions based on them. It needs to be rewritten after corrections in the Discussion and Conclusions sections.
Key-words
Terms used in the title should not be included as keywords because they are automatically indexed in the search engines. Please replace the terms that have been included in the title.
Material and Methods
Lines 110-120: as this evaluation was not part of the objectives of this research, nor did it relate to the animals or diets provided to the animals in this research, I suggest removing this part, as it is not possible to use these data for direct comparison.
Line 122: Please correct the beginning of the sentence after deleting the above paragraph.
Table 1 - 20:5 n3: Are FLAX and FO compositions correct?
Lines 194 and 198: Please use the same term or specify in line 198 that it was measured as rump fat thickness.
Line 199: I believe that GEST309 RFT was also measured, as stated in Table 3. If so, please include it in the text.
Line 213: Please specify the foal blood sample used for IgG analysis (pre-breastfeeding; 5 days?). If another collection time was used, please describe it (e.g. 12 h postpartum)..
Results
Lines 308-339: Food composition and food intake were not the objectives of this study; therefore, they should be presented in MM. I suggest simplifying this entire explanation on variations between diets, keeping only what can contribute to explaining variations obtained in the results.
Lines 312-314: I suggest excluding this, as it does not add relevant information and may raise questions about differences in diet.
Table 3: Since there were no statistical differences, the last row can be excluded from the subscripts of the table.
- Please include an explanation of the term "SEM" at the end of the table (check other tables that it must also be included).
Was the age of the mares similar between the treatments?
Line 347: Consider rewriting the sentence as follows: “Mean foal weights were not affected by the time versus treatment interaction (P = 0.6; Table 4), but were different among treatments (P = 0.03).
Lines 347-352: since the data are presented in Table 3, they should not be repeated in the text.
Table 4: Include SEM in the subscript of the table.
Line 357-365: if there was no difference in the parameter, it is more correct to present only the average value.
Lines 368-378: although these results are quite interesting and not so frequent to be found, they are not part of the same group of animals nor the same diet used in this experiment. In addition, the composition of the diet of these animals was not evaluated, making it unreasonable to present it as part of the study. It may be presented at a congress in the area.
Lines 383-405: since the data are presented in Table 3, they should not be repeated in the text.
Lines 422-425: This result should be reported in the abstract and considered in the conclusions.
Table 8: Align Columns
Discussion
Line 457: Please add 22:6 n-3 in parentheses after "DHA".
Line 502: Since there were no differences in terms of FAs in milk, it is not correct to say that one was more abundant than the other, because statistically, they were all the same.
Lines 542 and 543: There was a treatment effect on mare plasma regarding 20:2 n-6 (greater in FLAX mares). Redo the discussion of this part.
Conclusions
Please rewrite according to the changes made to the text.
Comments on the Quality of English LanguageThe English could be improved to more clearly express the research.
Author Response
This study compared the influence of supplementing flaxseed and fish oil at the end of gestation and peripartum on the plasma fatty acid profile of mares and foals and on mares’ milk composition. The current study provides a knowledge base for promoting the health of newborn foals through maternal nutrition.
Overall, the study was well designed, analysed, and written, and presented a very interesting discussion. Some attention is necessary to meet the journal’s standards in terms of tabulations and spacing between subtopics. Some statements were provided to enhance the quality of this study.
Response- we greatly appreciate the time and efforts of the reviewers to help us improve the manuscript.
The secondary objective lacks background information regarding the nutritional description and composition of diets, comprising a group of animals allocated to farms that apparently did not relate at all with the main data, making it difficult to justify the inclusion of these data, despite being potentially interesting. I suggest removing them from the article and correcting the related sections. It could perhaps be presented at some congress and indirectly cited throughout the discussion.
Response- data related to the secondary objective has been removed per the request of all three reviewers.
I take this opportunity to present some suggestions, which I believe would be of great value in the evaluation of foal health and could be considered as suggestions for further publications. It would be interesting to know whether there was any treatment effect on morbidity and/or mortality during the first week, first month, and until weaning. Would it be possible to retrieve this data for another study?
Response- None of the mares or foals suffered from morbidity or mortality during the first year of life. To our knowledge, all of the resultant foals are still alive and in productive careers.
SPECIFIC COMMENTS
Title
It should also include maternal parameters, as it was part of the study objective.
Response- mares’ milk added to the title
Simple Summary and Abstract
Corrections must be addressed by focusing on the objectives presented and the conclusions based on them. It needs to be rewritten after corrections in the Discussion and Conclusions sections.
Response- additions made to the abstract as requested. We are happy to make specific changes to the simple summary, however word limit requirements necessitate keeping it quite short.
Key-words
Terms used in the title should not be included as keywords because they are automatically indexed in the search engines. Please replace the terms that have been included in the title.
Response- keywords have been updated as requested.
Material and Methods
Lines 110-120: as this evaluation was not part of the objectives of this research, nor did it relate to the animals or diets provided to the animals in this research, I suggest removing this part, as it is not possible to use these data for direct comparison.
Response- removed as suggested
Line 122: Please correct the beginning of the sentence after deleting the above paragraph.
Response- Changed as suggested
Table 1 - 20:5 n3: Are FLAX and FO compositions correct?
Response- Yes these are correct.
Lines 194 and 198: Please use the same term or specify in line 198 that it was measured as rump fat thickness.
Response- This has been corrected as requested.
Line 199: I believe that GEST309 RFT was also measured, as stated in Table 3. If so, please include it in the text.
Response- We apologize for the lack of clarity, this has been corrected.
Line 213: Please specify the foal blood sample used for IgG analysis (pre-breastfeeding; 5 days?). If another collection time was used, please describe it (e.g. 12 h postpartum)..
Response- these blood samples were collected at 12 hours post partum. This information has been added.
Results
Lines 308-339: Food composition and food intake were not the objectives of this study; therefore, they should be presented in MM. I suggest simplifying this entire explanation on variations between diets, keeping only what can contribute to explaining variations obtained in the results.
Response- We agree that the macronutrients and DM were not meant to be different, however we did design the diets to cause a difference in DHA and ALA intake. Therefore, it felt like it belonged in the results. We greatly simplified the presentation of the results to focus on the most relevant pieces of data.
Lines 312-314: I suggest excluding this, as it does not add relevant information and may raise questions about differences in diet.
Response- This information has been removed per the suggestion of the reviewer.
Table 3: Since there were no statistical differences, the last row can be excluded from the subscripts of the table.
Response- we appreciate you catching this, it has been removed.
- Please include an explanation of the term "SEM" at the end of the table (check other tables that it must also be included).
Response- SEM definition has been added to each table where appropriate.
Was the age of the mares similar between the treatments?
Response- Algae mares averaged 19 years, Control mares averaged 16 years and Flax mares averaged 16.5 years respectively.
Line 347: Consider rewriting the sentence as follows: “Mean foal weights were not affected by the time versus treatment interaction (P = 0.6; Table 4), but were different among treatments (P = 0.03).
Response- this change has been made as requested.
Lines 347-352: since the data are presented in Table 3, they should not be repeated in the text.
Response- the text has been reduced so that it does not repeat the table quite as much.
Table 4: Include SEM in the subscript of the table.
Response- SEM has been included as a subscript in this and the other tables.
Line 357-365: if there was no difference in the parameter, it is more correct to present only the average value.
Response- this has been updated as requested.
Lines 368-378: although these results are quite interesting and not so frequent to be found, they are not part of the same group of animals nor the same diet used in this experiment. In addition, the composition of the diet of these animals was not evaluated, making it unreasonable to present it as part of the study. It may be presented at a congress in the area.
Response- these data have been removed from the current manuscript.
Lines 383-405: since the data are presented in Table 3, they should not be repeated in the text.
Response- The presentation of data in text has been reduced.
Lines 422-425: This result should be reported in the abstract and considered in the conclusions.
Response- An interpretation of this finding has been added to the discussion and the abstract.
Table 8: Align Columns
Response- aligned
Discussion
Line 457: Please add 22:6 n-3 in parentheses after "DHA".
Response- added as requested.
Line 502: Since there were no differences in terms of FAs in milk, it is not correct to say that one was more abundant than the other, because statistically, they were all the same.
Response- Thank you for this comment, these were not meant to be statistical comparisons. We did not compare percentages of fatty acids to each other, it was meant to be a broad observation. We removed the phrase, “regardless of treatment” to hopefully clarify that this was not meant to be about statistics or results of the study.
Lines 542 and 543: There was a treatment effect on mare plasma regarding 20:2 n-6 (greater in FLAX mares). Redo the discussion of this part.
Response- Thank you for pointing out this omission. We have added to the discussion as suggested.
Conclusions
Please rewrite according to the changes made to the text.
Response- changes made as requested.